# The Molecular Intersection of NEK1, C21ORF2, Cyclin F, and VCP in ALS Pathogenesis

**DOI:** 10.3390/genes16040407

**Published:** 2025-03-30

**Authors:** Yasuaki Watanabe, Tadashi Nakagawa, Makiko Nakagawa, Keiko Nakayama

**Affiliations:** 1Department of Neurology, Graduate School of Medicine, Tohoku University, Sendai 980-8575, Japan; yasuaki.watanabe.b8@tohoku.ac.jp; 2Division of Cell Proliferation, United Centers for Advanced Research and Translational Medicine, Graduate School of Medicine, Tohoku University, Sendai 980-8575, Japan; 3Department of Clinical Pharmacology, Faculty of Pharmaceutical Sciences, Sanyo-Onoda City University, Sanyo-Onoda 756-0084, Japan; 4Institute of Gene Research, Yamaguchi University Science Research Center, Ube 755-8505, Japan; 5Advanced Technology Institute, Life Science Division, Yamaguchi University, Yamaguchi 755-8611, Japan; 6Research Infrastructure Management Center, Institute of Science Tokyo, 1-5-45 Yushima, Bunkyo-ku, Tokyo 113-8510, Japan

**Keywords:** amyotrophic lateral sclerosis (ALS), TDP-43 aggregation, NEK1, C21ORF2, cyclin F, VCP, DNA damage repair, protein homeostasis, neurodegeneration

## Abstract

Amyotrophic lateral sclerosis (ALS) is a devastating neurodegenerative disorder characterized by the progressive degeneration of motor neurons, leading to muscle weakness, paralysis, and death. Although significant progress has been made in understanding ALS, its molecular mechanisms remain complex and multifactorial. This review explores the potential convergent mechanisms underlying ALS pathogenesis, focusing on the roles of key proteins including NEK1, C21ORF2, cyclin F, VCP, and TDP-43. Recent studies suggest that mutations in *C21ORF2* lead to the stabilization of NEK1, while *cyclin F* mutations activate VCP, resulting in TDP-43 aggregation. TDP-43 aggregation, a hallmark of ALS, impairs RNA processing and protein transport, both of which are essential for neuronal function. Furthermore, TDP-43 has emerged as a key player in DNA damage repair, translocating to DNA damage sites and recruiting repair proteins. Given that NEK1, VCP, and cyclin F are also involved in DNA repair, this review examines how these proteins may intersect to disrupt DNA damage repair mechanisms, contributing to ALS progression. Impaired DNA repair and protein homeostasis are suggested to be central downstream mechanisms in ALS pathogenesis. Ultimately, understanding the interplay between these pathways could offer novel insights into ALS and provide potential therapeutic targets. This review aims to highlight the emerging connections between protein aggregation, DNA damage repair, and cellular dysfunction in ALS, fostering a deeper understanding of its molecular basis and potential avenues for intervention.

## 1. Introduction

Amyotrophic lateral sclerosis (ALS) is a debilitating neurodegenerative condition marked by the progressive degeneration of both upper and lower motor neurons, ultimately leading to muscle weakness, paralysis, and mortality, primarily due to respiratory failure [1]. Advances in genetic analysis, applied to both familial ALS (fALS) and sporadic ALS (sALS), have enabled the identification of over 40 causative and risk-associated genes for ALS [2]. Functional studies of these genes, particularly those identified earlier, such as *SOD1*, *TARDBP* (which encodes TDP-43), *FUS*, and *C9ORF72*, combined with patient sample analysis and disease models, have provided essential insights into the molecular mechanisms underpinning ALS, encompassing the dysregulation of DNA, RNA, and proteins [3].

Given that abnormal protein aggregation is a hallmark of neurodegenerative disorders [4,5], research on protein misregulation surpasses that of DNA and RNA. For instance, mutations in genes such as *SOD1*, *TARDBP*, *FUS*, and *C9ORF72* have been shown to disrupt protein homeostasis by promoting protein aggregation. Mutant SOD1 forms cytotoxic aggregates that impair mitochondrial function [6,7,8], while hexanucleotide repeat expansions (HREs) in a non-coding region of *C9ORF72* generate dipeptide repeat proteins (DPRs) that self-assemble, sequestering RNA-binding proteins and proteins with low-complexity sequence domains (LCDs). This impairs the function of RNA-rich, membrane-less organelles such as nucleoli and stress granules, as well as nucleocytoplasmic transport mediated by proteins with LCDs [9,10,11]. Similarly, mutations in *TARDBP* and *FUS* increase the formation of cytoplasmic inclusions of TDP-43 and FUS [12,13,14,15], compromising several cellular activities, including protein nucleocytoplasmic transport [16], chaperoning [17], and degradation [18], by sequestering proteins essential for these critical processes.

The role of RNA misregulation in ALS has been suggested by the fact that several ALS-associated proteins, including TDP-43, FUS, and Matrin-3, are RNA-binding proteins [19]. Studies on these proteins have demonstrated that dysregulation of mRNA splicing, transport, and stability contributes to the degeneration of motor neurons [20]. Additionally, *C9ORF72* HREs generate toxic RNA foci that may be implicated in ALS pathogenesis [21,22,23,24].

In contrast to protein and RNA, evidence of DNA misregulation in ALS pathogenesis is relatively scarce. However, emerging research emphasizes DNA damage and defective DNA repair as crucial mechanisms in ALS [25], supported by postmortem studies revealing significant DNA damage in spinal motor neurons and the motor cortex [26,27]. These studies also highlight the activation of DNA damage response (DDR) pathways, including phosphorylated histone H2AX (γ-H2AX), phosphorylated ataxia telangiectasia mutated (p-ATM), and nuclear BRCA1, indicating persistent repair attempts [26,27,28]. TDP-43 and FUS regulate DDR and DNA repair through mechanisms such as non-homologous end joining (NHEJ), homologous recombination (HR), and base excision repair (BER) [25]. Thus, the pathological cytoplasmic mislocalization of TDP-43 and FUS in ALS disrupts nuclear DDR, leading to repair deficiencies [29,30]. *C9ORF72* HREs form G-quadruplexes and generate transcription-induced R-loops with RNA [31,32], destabilizing DNA and potentially increasing susceptibility to DNA damage [33]. Additionally, DPRs have been shown to inhibit NHEJ [34].

Despite significant progress over decades of research, the molecular mechanisms underlying ALS pathogenesis remain complex and multifactorial, prompting the question of how current knowledge converges or integrates to explain ALS pathogenesis. To address this challenge, investigating newly identified ALS-associated genes could provide valuable insights into the molecular pathogenesis from a novel perspective. Mutations in *NEK1*, *C21ORF2*, and *CCNF* (encoding cyclin F) in ALS patients were identified in 2016, and the functions ascribed to these genes—such as ciliogenesis and nuclear protein degradation—discussed below, were not closely linked to the established molecular mechanisms of ALS pathogenesis. Over the past 8–9 years, studies conducted by our group and others on these genes have uncovered two novel pathways—the C21ORF2-NEK1 and cyclin F-VCP pathways—in ALS pathogenesis, which enrich our understanding of ALS. This review summarizes the functions of these proteins and explores their potential contributions to a more comprehensive and unifying understanding of the molecular mechanisms underlying ALS. A literature search was conducted using the terms “NEK1”, “C21ORF2”, “CCNF”, “VCP”, and “ALS” in PubMed, with the references therein further analyzed to gather information on the molecular mechanisms of ALS pathogenesis associated with mutations in various genes, including *NEK1*, *C21ORF2*, *CCNF*, and *VCP*. To minimize bias, we prioritized studies cited in multiple sources or those containing original experimental data relevant to key themes of DNA damage repair, proteostasis, and TDP-43 pathology.

## 2. NEK1 and C21ORF2

### 2.1. Human Genetics

Mutations in the *NEK1* gene have been associated with a variety of disorders, including short-rib polydactyly syndrome [35], axial spondylometaphyseal dysplasia (SMD) [36], frontotemporal dementia (FTD) with ALS [37], and ALS [37,38,39,40]. A meta-analysis revealed that the prevalence of *NEK1* mutations in ALS patients is 3.1% with an odds ratio (OR) of 2.14 [41]. Likewise, mutations in *C21ORF2* (also referred to as *CFAP410* or *LRRC76*) have been identified in patients with SMD [42] and ALS [43]. Furthermore, retinal dystrophy [44,45] and asphyxiating thoracic dystrophy (also known as Jeune syndrome) [46] have also been linked to mutations in *C21ORF2*. To date, a meta-analysis specifically investigating the prevalence of *C21ORF2* mutations in ALS patients has not been conducted. However, the OR for *C21ORF2* mutation in ALS is estimated to be 1.45 [43].

### 2.2. Ciliogenesis

All diseases associated with mutations in *NEK1* or *C21ORF2*, with the exception of ALS, are classified as ciliopathies, where defects in ciliogenesis are thought to underlie the pathology [47]. Consequently, the role of *NEK1* and *C21ORF2* in ciliogenesis has been extensively studied (Figure 1). Gene knockout (KO) of either *NEK1* or *C21ORF2* in human untransformed retinal pigment epithelial cells results in a significant reduction in ciliogenesis, suggesting that both NEK1 and C21ORF2 are essential for ciliogenesis [48]. At the molecular level, NEK1 and C21ORF2 form a stable complex, mediated by a C-terminal acidic domain in NEK1 (aa 1160–1286), referred to as the C21ORF2 interaction domain (CID) [48], which facilitates mutual stabilization of the proteins [48,49] (Figure 1).

Pathogenic point mutations in C21ORF2 (R73P and L224P) or in the CID of NEK1 (D1277A) weaken this interaction [48], indicating that complex formation is crucial for their functional roles. Consistently, the NEK1 D1277A mutant is unable to fully rescue ciliogenesis defects in *NEK1* KO cells, and C21ORF2 mutants (R73P and L224P) fail to completely restore ciliogenesis in *C21ORF2* KO cells. Centrosomes act as nucleation sites for cilia, and both NEK1 and C21ORF2 are localized to these structures [50,51]. ALS-associated C21ORF2 mutants (V58L, R106C, R172W, A255T) exhibit reduced localization to centrosomes and diminished ability to rescue ciliogenesis defects in human neuroblastoma SH-SY5Y cells [52], suggesting that these proteins function at the centrosome to initiate ciliogenesis. However, the precise molecular mechanisms by which NEK1 and C21ORF2 regulate ciliogenesis remain to be fully elucidated. Notably, overexpression of NEK1 also inhibits ciliogenesis [51], underscoring the importance of balanced activity in NEK1 for the regulation of ciliogenesis.

### 2.3. DNA Repair

NEK1 has been identified as playing a critical role in DNA damage repair (Figure 1), a vital defense mechanism for the survival of post-mitotic neurons [53]. The accumulation of DNA damage with aging may serve as a central pathological driver in ALS progression [54]. Upon exposure to DNA-damaging ionizing radiation, NEK1 rapidly localizes to DNA damage sites, positioning itself as an early responder in the DNA damage response (DDR) pathway [55]. NEK1 interacts with key DDR components, including ataxia telangiectasia and Rad3-related (ATR) and ATR-interacting protein (ATRIP), thereby priming ATR for efficient DNA damage signaling, which is essential for activation of downstream repair pathways [56].

In addition to its role in DDR signaling, NEK1 directly participates in both NHEJ and HR DNA repair by regulating critical repair factors. NEK1 interacts with Ku80, a key component of the NHEJ pathway, facilitating the loading of replication factors onto chromatin and promoting S-phase progression during DNA replication and repair [57]. NEK1 also regulates RAD54, an essential factor in HR, thus orchestrating HR repair and ensuring replication fork stability [58]. Consistent with this, loss of NEK1 function severely impairs DNA repair capacity, resulting in delayed repair kinetics and failure to properly arrest the cell cycle following DNA damage [59], thereby increasing sensitivity to DNA-damaging agents [60]. NEK1 also increases the expression of DNA repair pathway genes [61]. These findings underscore NEK1 as a multifunctional kinase that integrates DNA damage recognition, repair coordination, and gene regulation to maintain genomic integrity.

Studies involving ALS patient-derived motor neurons have shown that loss of NEK1 function leads to a significant accumulation of DNA damage. Specifically, *NEK1* loss-of-function (LoF) mutations result in increased levels of γH2AX foci, a marker of DNA double-strand breaks, and reduced DNA repair capacity in motor neurons [62]. The DNA damage response pathway is notably compromised in these cells, with impaired recruitment of DNA repair proteins to damage sites. Furthermore, in iPSC-derived motor neurons carrying both the HREs in *C9ORF72* and a LoF mutation in *NEK1*, an increase in RNA foci and exacerbation of DNA damage were observed compared to *C9ORF72* single mutants. These findings suggest that NEK1 dysfunction exacerbates DNA damage in *C9ORF72*-mutant motor neurons, potentially acting as a genetic modifier in ALS pathology [63]. The accumulation of DNA damage in motor neurons due to NEK1 dysfunction represents a potential mechanism contributing to motor neuron degeneration in ALS, emphasizing the importance of proper DNA repair mechanisms in maintaining motor neuron health and survival.

C21ORF2 is also implicated in DNA repair processes (Figure 1), as depletion of C21ORF2 reduces the efficiency of HR repair of damaged DNA, which can be rescued by NEK1 overexpression [48,64]. These results suggest that C21ORF2 functions within the same pathway as NEK1 in DNA damage repair, as it does in ciliogenesis. However, in contrast to NEK1, C21ORF2 does not translocate to damaged DNA sites [64], indicating its supportive, rather than essential, role in NEK1 function in DNA repair processes. Recent studies have highlighted that primary cilia play a crucial role in maintaining genome stability by regulating DNA repair processes in cholangiocytes [65]. For instance, DNA repair proteins such as RAD51, ATR, PARP1, CHK1, and CHK2 have been found to colocalize with primary cilia [65]. Additionally, deciliated cells exhibit downregulation of key DDR proteins, such as ATM, p53, and p21, following irradiation, further emphasizing the role of primary cilia in maintaining genomic integrity [65]. Consistently, deciliated cells exhibit reduced survival, increased S-phase arrest, and enhanced DNA damage when exposed to genotoxic agents [65]. While these data suggest that ciliogenesis defects contribute to defective DDR and DNA damage repair, the necessity of primary cilia for DDR and DNA damage repair, along with the detailed mechanisms involved, requires further elucidation in motor neurons.

### 2.4. Protein Homeostasis

NEK1 is a pivotal kinase that regulates protein stability through the phosphorylation of target proteins (Figure 1). Recent studies have uncovered distinct regulatory mechanisms by which NEK1 influences various substrates. NEK1 directly phosphorylates the Von Hippel–Lindau (VHL) tumor suppressor, promoting its proteasome-dependent degradation [66]. This phosphorylation-induced destabilization of VHL has been shown to affect ciliogenesis and maintenance [66]. In contrast, NEK1 exhibits different regulatory effects on C21ORF2, as described in Section 2.1. NEK1-mediated hyperphosphorylation of C21ORF2 prevents its interaction with the ubiquitin ligase F-box only protein 3 (FBXO3), thereby stabilizing C21ORF2 [49]. Notably, the ALS-associated C21ORF2 mutant (V58L) displays heightened susceptibility to NEK1-mediated hyperphosphorylation, suggesting a potential role in ALS pathogenesis [49]. These findings illustrate the bidirectional regulatory capacity of NEK1, which can either promote protein degradation, as seen with VHL, or enhance protein stability, as observed with C21ORF2. This dual regulatory mechanism underscores NEK1’s significance in fine-tuning various cellular functions through the differential control of protein stability. The contrasting effects of NEK1-mediated phosphorylation on different substrates highlight its complex role in cellular homeostasis and disease pathogenesis.

NEK1 is also essential for global protein homeostasis through its interactions with key regulators, including heat shock proteins (HSPA1A, HSPA8, HSP90A1, and HSP90AB1) and ubiquitin [67]. These proteins govern protein folding, repair, and degradation—processes closely linked to neurodegenerative diseases such as ALS. Loss of NEK1 disrupts these pathways, leading to the accumulation of misfolded proteins and impaired clearance [67]. Proteomic analysis of NEK1-deficient motor neurons has revealed significant changes in protein expression, reinforcing its role in proteostasis [67]. Given that protein aggregation and impaired degradation are hallmarks of ALS, NEK1 dysfunction may contribute to disease progression by destabilizing proteostasis networks.

Dysregulation of NEK1-mediated proteostasis may extend beyond its effects on other proteins to impact NEK1 itself, potentially contributing to ALS pathogenesis. Although the evidence is limited, emerging data suggest that in addition to NEK1 dysfunction, increased NEK1 activity or accumulation may also play a role in ALS. While nonsense mutations in *NEK1* primarily support a LoF mechanism, the effects of missense mutations, such as R261H, on NEK1 kinase activity and protein expression remain unclear [38,39]. NEK1 has been shown to phosphorylate C21ORF2, preventing its degradation by the FBXO3-mediated ubiquitin-proteasome system (UPS), as described above. This phosphorylation-induced stabilization of C21ORF2, in turn, enhances NEK1 stability, creating a self-reinforcing regulatory loop in which NEK1 indirectly regulates its own protein levels [49]. Disruption of this cycle, whether through LoF or abnormal stabilization, could contribute to ALS pathogenesis. Consistently, the ALS-associated C21ORF2-V58L variant has been suggested to abnormally stabilize NEK1, leading to its increased accumulation [49]. Notably, in the motor cortex of an ALS patient carrying the R261H missense mutation, cytoplasmic NEK1 protein accumulation has been observed, along with TDP-43 aggregation [68]. These findings raise critical questions about the regulation of NEK1 protein levels in ALS pathogenesis, warranting further investigation.

## 3. Cyclin F and VCP

### 3.1. Human Genetics

Point mutations in the exonic regions of the cyclin F gene (*CCNF*) have been identified in both familial and sporadic ALS patients, with or without frontotemporal dementia (FTD) [69]. All *CCNF* mutations in these patients lead to amino acid substitutions, with no deletions or protein-disrupting mutations detected, suggesting a gain-of-function (GoF) mechanism that is toxic to cells [69]. To date, no other diseases associated with *CCNF* mutations have been identified, although elevated cyclin F expression levels have been correlated with poorer prognosis in patients with hepatocellular carcinoma [70], clear cell renal cell carcinoma [71], ovarian cancer [72], and breast cancer [73]. The frequency of *CCNF* mutations in ALS patients is estimated to range from 0.6% to 3.3% [69] with effect sizes from 0.005 to 0.033 [74]. *VCP* mutations have been implicated in multisystem proteinopathy 1 (MSP1) (also known as inclusion body myopathy associated with Paget disease of bone and FTD) [75,76], Charcot-Marie-Tooth type 2 disease [77], Parkinson’s disease (PD) [78,79], hereditary spastic paraplegia [80], and ALS [81], with the latter condition added to the list in 2010. Mutations in *VCP* do not result in deletions, indicating a GoF mechanism, similar to that of *CCNF* mutations. Studies on patients with *VCP* mutations report that myopathy, Paget’s disease of bone, and FTD were present in 90%, 42%, and 30% of the patients, respectively, while approximately 9% of patients exhibited an ALS phenotype and 4% were diagnosed with PD [76]. The frequency of *VCP* mutations has been reported to be 0.28% in patients with fALS and 0.06% in patients with sALS [82].

Comprehensive and up-to-date overviews of cyclin F [83] and VCP [84] in the context of ALS pathogenesis have recently been provided. Therefore, this review centers on our findings and their related studies.

### 3.2. VCP Activation by Cyciln F

To elucidate the molecular mechanisms underlying ALS pathogenesis, we investigated the potential interaction of cyclin F with other proteins implicated in ALS and discovered that wild-type cyclin F binds to p62 (also known as SQSTM1), TDP-43, and VCP, with the strongest binding observed to VCP [85]. VCP functions as a molecular chaperone and segregase, utilizing ATP hydrolysis to extract proteins from large cellular structures for degradation or recycling, thereby maintaining protein homeostasis and supporting various cellular processes [86,87]. We found that the N-terminal region of cyclin F is responsible for its interaction with VCP (Figure 2), and mutations identified in ALS patients enhance this binding. In parallel with this binding, the ATPase activity of VCP is significantly increased [85]. Furthermore, some of the mutations mislocalize cyclin F to the cytoplasm, whereas the wild-type protein predominantly resides in the nucleus [85].

As described earlier, *VCP* itself is subject to point mutations in ALS [81], some of which have been shown to augment its ATPase activity [88,89]. Given that VCP is reported to possess unfoldase activity mediated by its ATPase function [90] and that protein aggregation, frequently observed in ALS patient motor neurons, is thought to be driven by unstructured regions susceptible to unfolding [91], we hypothesized that cyclin F mutants, which activate VCP’s ATPase activity, lead to excessive unfoldase activity of VCP, resulting in abnormal protein unfolding and subsequent aggregation. Consistent with this hypothesis, we demonstrated that increased VCP ATPase activity promotes oxidative stress-induced TDP-43 cytoplasmic aggregation [85], a characteristic observed in approximately 90% of ALS patients [92]. In support of our findings, a recent study highlighted the critical role of molecular chaperones in TDP-43 folding, which prevents the formation of TDP-43 aggregates [93,94,95]. Furthermore, recent experiments in which wild-type and mutant cyclin F were overexpressed in the mouse brain demonstrated that mutant cyclin F promotes TDP-43 aggregation with colocalization of VCP [96], further corroborating our results.

Although cyclin F is a prototypical F-box protein that functions as a substrate receptor in the Cullin1-RING ubiquitin ligase complex (CRL1, also known as SCF) [97], we did not observe changes in the ubiquitylation levels of VCP in cells overexpressing cyclin F [85], despite extensive ubiquitylation of the known substrate, RRM2, in the same experimental condition [85,98]. Therefore, ubiquitylation likely does not contribute to cyclin F-mediated activation of VCP ATPase activity, revealing a novel function of cyclin F.

### 3.3. Cyclin F as an Ubiquitin Ligase for ALS-Associated Proteins

While our study primarily focused on VCP among the cyclin F binding proteins, other research has investigated the direct interaction between cyclin F and TDP-43, reporting that cyclin F directly binds to TDP-43 and facilitates CRL1^Cyclin F^-mediated ubiquitylation, leading to proteasomal degradation [99,100]. Interestingly, cyclin F mutants derived from ALS patients were shown to enhance K48-linked poly ubiquitylation, a well-established marker for proteasomal degradation [101], yet instead of promoting degradation, these mutants result in protein aggregation [99]. This discrepancy warrants further investigation. Additionally, while these findings are intriguing, they require validation under more physiological conditions, particularly since the data were primarily derived from experiments involving overexpressed cyclin F and TDP-43. Notably, cyclin F is an unstable protein, whereas TDP-43 is relatively stable, suggesting that the effects of cyclin F on TDP-43 may be transient. Furthermore, the overexpression of cyclin F could introduce experimental artifacts.

In addition to TDP-43, p62 has also been reported to undergo ubiquitylation by CRL1^cyclin F^, promoting p62 aggregation [102]. Compared to wild-type cyclin F, the ALS patient-derived cyclin F mutant exhibits a greater ability to enhance p62 ubiquitylation but unexpectedly reduces p62 aggregation [102]. Thus, no direct correlation between ubiquitylation and p62 aggregation is evident. Similar to TDP-43, this study primarily relied on protein overexpression, and, therefore, the results require validation in more physiologically relevant contexts. Importantly, p62 has been shown to enhance TDP-43 aggregation [103], suggesting that cyclin F, VCP, and p62 may act synergistically to promote the formation of pathogenic TDP-43 aggregates.

## 4. Possible Convergent Mechanisms of ALS

The discussions in the preceding sections suggest that mutations in *C21ORF2*, *NEK1*, *cyclin F*, and *VCP* play central roles in ALS pathogenesis. C21ORF2 mutant-induced stabilization of NEK1, together with cyclin F mutant-induced VCP activation and the subsequent aggregation of TDP-43, provides compelling evidence that multiple pathways contribute to ALS. However, the question remains whether these mechanisms operate independently or converge at a critical juncture in the disease process. TDP-43 aggregation is a hallmark of ALS, and its presence has been noted in ALS patients with *NEK1* mutations [68]. This observation raises the possibility that TDP-43 aggregation might occur at or downstream of an intersection between these pathways. Despite such evidence, the exact mechanism by which *NEK1* mutations induce TDP-43 aggregation remains unclear, suggesting the need for further investigation to uncover whether these processes are truly independent or intertwined.

The cyclin F T31 phosphorylation site, which is catalyzed by AKT [104], lies within a preferred sequence for NEK1 phosphorylation [105], suggesting the intersection between cyclin F and NEK1. NEK1’s potential role in phosphorylating this site may serve to stabilize cyclin F, enhancing VCP ATPase activity and promoting subsequent TDP-43 aggregation, which is a central pathological feature in ALS. TDP-43 aggregation is now understood to mediate ALS pathogenesis through both LoF and GoF mechanisms [106,107]. The aggregation of TDP-43 impairs its normal role in regulating mRNA splicing, a critical function for the maintenance of cellular homeostasis. Additionally, it leads to the sequestration of essential proteins that govern proteostasis and protein transport [108,109]. The disruption of these cellular processes is one of the core drivers of ALS progression. Therefore, understanding the precise factors contributing to TDP-43 aggregation—such as the potential convergence of NEK1 and cyclin F signaling—could provide valuable insights into ALS pathogenesis and novel therapeutic targets.

In addition to its role in mRNA processing, emerging evidence supports a critical role for TDP-43 in DNA damage repair (Figure 3). TDP-43 translocates to DNA damage sites upon double-strand breaks, where it recruits DNA ligase to seal the breaks, facilitating the repair process [30]. In instances of DNA mismatch, TDP-43 interacts with mismatch repair proteins such as MLH1 and MSH6, which are crucial for efficient repair [110]. Furthermore, TDP-43 has been shown to regulate the expression of these mismatch repair proteins even in the absence of DNA-damaging stress, suggesting that TDP-43 plays an important role in enhancing cellular resistance to DNA damage [111]. This adds another layer of complexity to the role of TDP-43 in ALS, as disruption of DNA repair mechanisms may contribute to the pathogenesis of the disease. The pathological aggregation of TDP-43 in the cytoplasm disrupts nuclear transport by displacing transport factors from their proper localization, as described earlier [108]. This disruption affects the nuclear translocation of proteins involved in DNA damage repair, thereby potentially compromising genomic integrity. The ALS-associated TDP-43 Q331K mutant, which exhibits reduced nuclear localization and increased cytoplasmic mislocalization, impairs the nuclear import of XRCC4-DNA ligase 4, a critical component of the NHEJ pathway [112]. Furthermore, ALS/FTD patients with TDP-43 pathology exhibit an accumulation of the DNA damage markers, further supporting the hypothesis that TDP-43 dysfunction leads to impaired DNA repair and genomic instability [26,27,28]. Collectively, these findings suggest that defects in TDP-43-mediated DNA repair mechanisms contribute to disease pathogenesis, potentially exacerbating neuronal vulnerability in ALS.

Interestingly, in addition to NEK1/C21ORF2 module, cyclin F and VCP have also been implicated in DNA damage repair. Cyclin F, as part of the CRL1 ubiquitin ligase complex, targets atypical E2F transcription factors (E2F7 and E2F8) for proteasomal degradation [113]. This degradation is essential for the expression of DNA repair genes, which are involved in various DNA repair pathways [113]. Cyclin F also contributes to maintaining genome integrity by regulating degradation of RRM2, a regulatory component of dNTP-generating ribonucleotide reductase complex [98]. Cyclin F is downregulated in response to DNA damage to allow accumulation of RRM2 and support DNA damage repair [98]. VCP is involved in the mobilization of damage sensors to repair executors by extracting these sensors from chromatin using its ATPase activity [114,115]. This mechanism is essential for nucleotide excision repair and double-strand break repair [114,115]. Given the essential role of VCP in mobilizing repair proteins, it is likely that mutations in *VCP* or alterations in its regulation by cyclin F could lead to impaired DNA repair and contribute to ALS pathogenesis (Figure 3).

Considering the substantial accumulation of DNA damage observed in the motor neurons of ALS patients [26,27,28], it is plausible that impaired DNA damage repair represents a common downstream mechanism in ALS pathogenesis. Mutations in *C21ORF2*, *NEK1*, *cyclin F*, and *VCP* compromise DNA repair pathways, leading to genomic instability. As DNA repair mechanisms are disrupted, the accumulation of genome mutations might cause the accumulation of abnormal proteins, which may disrupt the protein homeostasis and accelerate neurodegeneration, contributing to the onset and progression of ALS (Figure 3). Therefore, understanding how these mutations converge to impair DNA repair, protein homeostasis, and TDP-43 function could provide critical insights into the molecular underpinnings of ALS and open new avenues for therapeutic intervention.

Currently, no specific modulators of C21ORF2 and cyclin F are available, while VCP inhibitors and NEK1 inhibitors are accessible. ML240, a VCP ATPase inhibitor, has been shown to reduce DNA damage, as well as abnormal splicing and mislocalization of RNA and proteins, in iPS cell-derived motor neurons with either *TARDBP* or *VCP* mutations [116,117]. Similarly, CB-5083 and NMS-873, also VCP inhibitors, have demonstrated the ability to rescue the survival of motor neurons differentiated from iPS cells with *VCP* mutations [118]. While the efficacy of these VCP inhibitors in ALS model mice has not yet been evaluated, these findings provide a rationale for targeting VCP to ameliorate ALS pathogenesis. The efficacy of NEK1 inhibitors has not been tested in the context of ALS [119,120], but their development is eagerly anticipated for targeting the NEK1/C21ORF2 pathway.

Although genetic mutations serve as biomarkers for identifying individuals at risk for ALS, they cannot predict disease progression or its stages. A recent finding that elevated TDP-43 levels in plasma extracellular vesicles distinguish ALS from other neurodegenerative disorders with high diagnostic accuracy [121] suggests the potential use of ALS-related proteins in plasma as biomarkers for assessing the state of ALS, although further studies are required to validate these biomarkers in preclinical stages. As discussed in Section 2.4, an ALS patient with a *NEK1* mutation exhibits cytoplasmic NEK1 protein accumulation [68]. Additionally, mutant cyclin F identified in ALS patients frequently mislocalizes to the cytoplasm [85]. These results suggest that NEK1, C21ORF2 (which is stabilized by NEK1) [48,49], and cyclin F might be elevated in the plasma of ALS patients with mutations in these genes, particularly when the motor neurons of these patients undergo cell death, and thus could serve as biomarkers for patient-specific ALS progression. Research aimed at exploring these possibilities would be valuable for further classifying ALS patients and developing tailored, patient-specific therapies.

## Figures and Tables

**Figure 1 genes-16-00407-f001:**
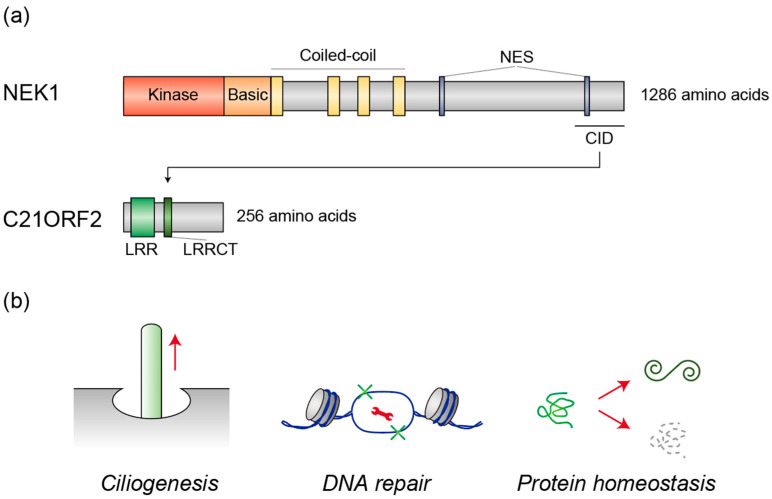
(**a**) Schematic depiction of the domain structures of NEK1 and C21ORF2. NEK1 comprises kinase, basic, and coiled-coil domains, along with nuclear export signals (NES), while C21ORF2 contains leucine-rich repeats (LRR) and an LRR C-terminal flanking region (LRRCT). The C21ORF2-interacting domain (CID) in NEK1 facilitates its interaction with C21ORF2. (**b**) Cellular processes regulated by the NEK1/C21ORF2 complex. The NEK1/C21ORF2 complex plays a pivotal role in ciliogenesis, DNA repair, and protein homeostasis. The wrench represents proteins that are involved in DNA repair processes.

**Figure 2 genes-16-00407-f002:**
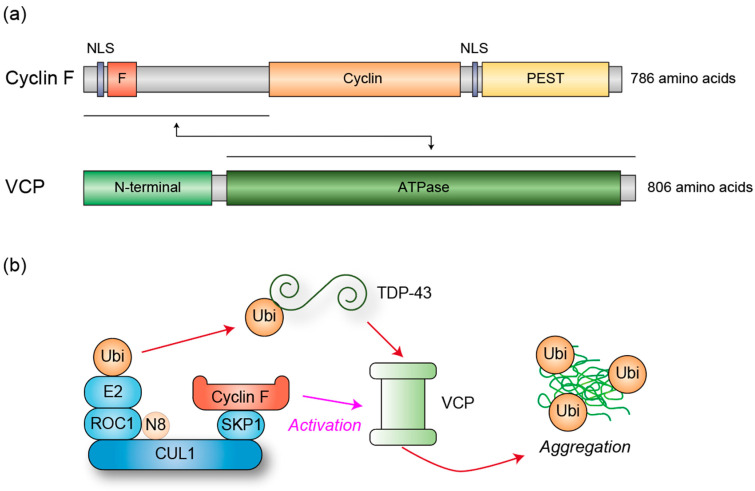
(**a**) Schematic representation of the domain structures of cyclin F and VCP. Cyclin F consists of F-box (F), cyclin, and PEST domains, along with nuclear localization signals (NLS), whereas VCP contains N-terminal and ATPase domains. The N-terminal region of cyclin F interacts with the ATPase domain of VCP. (**b**) Molecular mechanism underlying TDP-43 aggregation regulated by cyclin F and VCP. Cyclin F binds to TDP-43 to facilitate its ubiquitylation by the CUL1 complex. Concurrently, cyclin F associates with VCP to activate its ATPase activity. VCP induces a conformational change in TDP-43, thereby promoting its aggregation. VCP is known to recognize ubiquitylated proteins, suggesting that TDP-43 ubiquitylation likely precedes recognition by VCP, although it is also possible that TDP-43 is ubiquitylated subsequent to VCP-mediated unfolding.

**Figure 3 genes-16-00407-f003:**
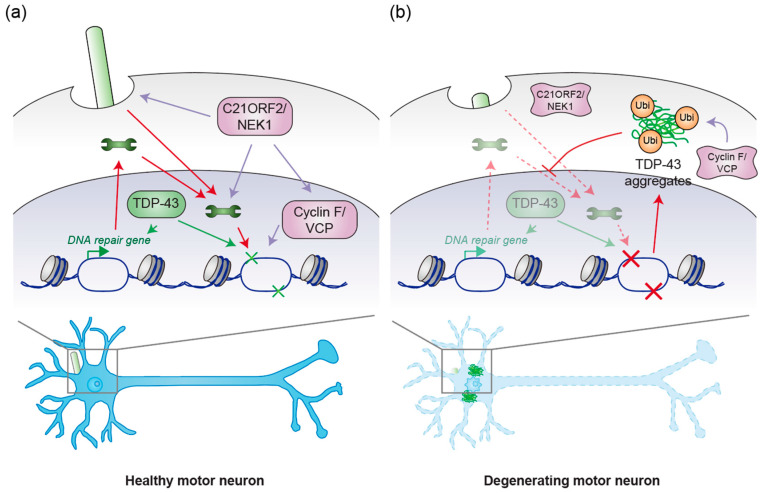
(**a**) Potential intersection of the C21ORF2/NEK1 module, the cyclin F/VCP module, and TDP-43 in healthy motor neurons. The C21ORF2/NEK1 module facilitates ciliogenesis, which has been demonstrated to stabilize proteins involved in DNA repair. Additionally, C21ORF2/NEK1 plays a pivotal role in the activity of DNA repair-associated proteins. NEK1 may also stabilize cyclin F by phosphorylating T31 residue. VCP, whose activity is supported by cyclin F, also contributes to DNA repair processes. Furthermore, TDP-43 is implicated in the regulation of genes associated with DNA repair, as well as in the DNA repair mechanisms themselves. (**b**) Potential intersection of mutant C21ORF2/NEK1, mutant cyclin F/VCP, and TDP-43 aggregation in the pathogenesis of ALS. Mutations in *C21ORF2* affect the protein levels of NEK1 and vice versa, resulting in abnormal NEK1 kinase activity, defects in ciliogenesis, and impairments in DNA repair processes. Mutant cyclin F, often associated with cytoplasmic translocation, aberrantly activates VCP ATPase activity. This activation, also induced by *VCP* mutations, promotes the aggregation of TDP-43, as illustrated in Figure 2. These aggregates disrupt nuclear protein import by sequestering proteins involved in nucleocytoplasmic transport, thereby potentially decreasing the concentration of DNA repair-related proteins in the nucleus and impairing cellular DNA repair activity. Simultaneously, this disrupts the ability to enhance the expression of DNA repair genes and the DNA repair processes themselves, due to the reduction in nuclear TDP-43. The failure to repair damaged DNA may lead to genomic mutations that perturb protein homeostasis, promoting the further accumulation of TDP-43 aggregates. The wrench represents the protein that is involved in DNA repair processes.

## Data Availability

No new data were created or analyzed in this study. Data sharing is not applicable to this article.

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
