# Peer review of "The Molecular Intersection of NEK1, C21ORF2, Cyclin F, and VCP in ALS Pathogenesis"

_genes, 2025, doi:10.3390/genes16040407_

Round 1
Reviewer 1 Report
Comments and Suggestions for Authors
Authors reviewed the pivotal genes/proteins in the pathogenesis of ALS, and they covered well for each gene/proteins in their respective pathways.
It will be good for author to draw overall schematic figure of connecting above gene/proteins in global pathogenesis towards ALS.
Authors should also include developing thrapeutics targetting each pathogenic pathway.
It will be interesting to include ALS biomarkers in relation with each pathway.
Author Response
We sincerely appreciate your thorough review of our manuscript. Your comments have been invaluable in enhancing the clarity and quality of our work. Below, we provide detailed responses and outline the corresponding revisions, which are highlighted in the resubmitted files.
Comments: Authors reviewed the pivotal genes/proteins in the pathogenesis of ALS, and they covered well for each gene/proteins in their respective pathways.
Response: Thank you for the positive comment.
Comments: It will be good for author to draw overall schematic figure of connecting above gene/proteins in global pathogenesis towards ALS.
Response: We have added healthy and degenerating motor neurons in the summary figure to illustrate the functions of the proteins discussed in this review in the broader context of ALS pathogenesis (Figure 3).
Comment: Authors should also include developing thrapeutics targetting each pathogenic pathway.
Response: We have included developing therapeutic targeting for each pathogenic pathway, as shown below (line 417-427).
Added sentences: “Currently, no specific modulators of C21ORF2, and Cyclin F are available, while VCP inhibitors and NEK1 inhibitors are accessible. ML240, a VCP ATPase inhibitor, has been shown to reduce DNA damage, as well as abnormal splicing and mislocalization of RNA and proteins, in iPS cell-derived motor neurons with either TARDBP or VCP mutations [116, 117]. Similarly, CB-5083 and NMS-873, also VCP inhibitors, have demonstrated the ability to rescue the survival of motor neurons differentiated from iPS cells with VCP mu-tations [118]. While the efficacy of these VCP inhibitors in ALS model mice has not yet been evaluated, these findings provide a rationale for targeting VCP to ameliorate ALS pathogenesis. The efficacy of NEK1 inhibitors has not been tested in the context of ALS [119, 120], but their development is eagerly anticipated for targeting the NEK1/C21ORF2 pathway.”
Comment: It will be interesting to include ALS biomarkers in relation with each pathway.
Response: We have included ALS biomarkers in relation with each pathway as below (line 428-441).
Added sentences: “Although genetic mutations serve as biomarkers for identifying individuals at risk for ALS, they cannot predict disease progression or its stages. A recent finding that elevated TDP-43 levels in plasma extracellular vesicles distinguish ALS from other neurodegenerative disorders with high diagnostic accuracy [121] suggests the potential use of ALS-related proteins in plasma as biomarkers for assessing the state of ALS, although further studies are required to validate these biomarkers in preclinical stages. As discussed in Section 2.4, an ALS patient with a NEK1 mutation exhibits cytoplasmic NEK1 protein accumulation [68]. Additionally, mutant Cyclin F identified in ALS patients frequently mislocalizes to the cytoplasm [85]. These results suggest that NEK1, C21ORF2 (which is stabilized by NEK1) [48, 49], and Cyclin F might be elevated in the plasma of ALS patients with mutations in these genes, particularly when the motor neurons of these patients undergo cell death, and thus could serve as biomarkers for patient-specific ALS progression. Research aimed at exploring these possibilities would be valuable for further classifying ALS patients and developing tailored, patient-specific therapies.”
Reviewer 2 Report
Comments and Suggestions for Authors
The authors complete a review of human ALS with emphasis on the roles of NEK1, 19 C21ORF2, Cyclin F, VCP, and TDP-43. While the review is rather narrowly defined in scope, the review does a comprehensive of evaluating these designated targets. The literature is diverse and good pictorial depictions of mechanisms further support the readability. A few minor changes to consider:
The authors should do a better job justifying the scope of the review. The role of the primary targets is justified, but how the authors arrived at this list is not well justified.
The authors did well with qualitative descriptions. However, quantitative data would make better arguments. For example, citing data that provides odds ratio, effect sizes, etc. to better quantitatively describe the role and impact of the review's targets.
The authors do not really describe how they arrived at the papers included in this review. While this is not listed as a systematic review, it would still benefit with a brief description on how literature was searched and selected and the overall scope.
Author Response
We sincerely thank you for your constructive comments, which have greatly contributed to the refinement of our manuscript. Detailed responses to your suggestions are provided below, with corresponding revisions highlighted in the resubmitted files.
Comment: The authors complete a review of human ALS with emphasis on the roles of NEK1, C21ORF2, Cyclin F, VCP, and TDP-43. While the review is rather narrowly defined in scope, the review does a comprehensive of evaluating these designated targets. The literature is diverse and good pictorial depictions of mechanisms further support the readability. A few minor changes to consider:
Response: Thank you for the positive comment.
Comment: The authors should do a better job justifying the scope of the review. The role of the primary targets is justified, but how the authors arrived at this list is not well justified.
Response: We have added the following sentence in the introduction (line 85-89) to justify the selection of study targets in relation to ALS pathogenesis.
Previous manuscript: “Mutations in NEK1, C21ORF2, and CCNF (encoding Cyclin F) in ALS patients were identified in 2016, and studies conducted by our group and others on these genes have uncov-ered two novel pathways—the C21ORF2-NEK1 and Cyclin F-VCP pathways—in ALS pathogenesis.”
Revised manuscript: “Mutations in NEK1, C21ORF2, and CCNF (encoding Cyclin F) in ALS patients were identified in 2016, and the functions ascribed to these genes—such as ciliogenesis and nuclear protein degradation—discussed below, were not closely linked to the established molecular mechanisms of ALS pathogenesis. Over the past 8–9 years, studies conducted by our group and others…”
Comment: The authors did well with qualitative descriptions. However, quantitative data would make better arguments. For example, citing data that provides odds ratio, effect sizes, etc. to better quantitatively describe the role and impact of the review's targets.
Response: We have added the odds ratio or effect sizes associated with NEK1 (line 105-106), C21ORF (line 111), and CCNF mutation (line 248-249) in ALS patients. To the best of our knowledge, specific odds ratios or effect sizes for ALS risk between individuals with VCP mutations and those without are not directly provided in the available literature. Instead, we have added the frequency of VCP mutations in ALS patients (line 257-258).
Comment: The authors do not really describe how they arrived at the papers included in this review. While this is not listed as a systematic review, it would still benefit with a brief description on how literature was searched and selected and the overall scope.
Response: We have added a brief description of how the literature was searched, selected, and the overall scope, as shown below (lines 94-99).
Added sentences: “A literature search was conducted using the terms “NEK1,” “C21ORF2,” “CCNF,” “VCP,” and “ALS” in PubMed, with the references therein further analyzed to gather information on the molecular mechanisms of ALS pathogenesis associated with mutations in various genes, including NEK1, C21ORF2, CCNF, and VCP. To minimize bias, we prioritized studies cited in multiple sources or those containing original experimental data relevant to key themes of DNA damage repair, proteostasis, and TDP-43 pathology.”